# OpenReview forum: "PICACO: Pluralistic In-Context Value Alignment via Total Correlation Optimization"
_ICLR.cc/2026/Conference — Submitted to ICLR 2026_

### Official Review · Reviewer_kUR8 · 2025-10-27

**Soundness:** 2
**Presentation:** 3
**Contribution:** 3
**Rating:** 8
**Confidence:** 3

**Summary:**

The paper proposes a novel in-context learning method named PICACO, which aims to address the challenge faced by LLMs in adhering to multiple, even conflicting, human value systems. PICACO optimizes a meta-instruction to maximize the total correlation between the model’s responses and a target set of human values, thereby achieving more balanced and effective multi-value alignment without fine-tuning the underlying model parameters. Experimental results demonstrate that PICACO consistently outperforms existing ICA baselines across diverse value compositions and different LLM architectures, achieving superior steerability, robustness, and value conformity.

**Strengths:**

1. The definition of the multi-value alignment problem is novel and of significant importance.
2. The introduction of information theory concepts into the in-context alignment task is not only intuitively reasonable but also provides a solid mathematical foundation for the optimization objective, surpassing simple heuristic methods.
3. The proposed method is effective for both black-box models and open-source models.

**Weaknesses:**

1. Although the proposed method demonstrates some empirical ability to mitigate value conflicts, it lacks a theoretical framework for modeling the intrinsic nature of such conflicts. This limitation constrains its reliability and interpretability in extreme or complex ethical scenarios.

2. PICACO depends heavily on an external model (e.g., *GPT-4o-mini*) as the value evaluator $q_{\omega}$, which may introduce evaluation bias.

**Questions:**

1.What are the computational and sampling costs of PICACO?

2.How can the PICACO framework be extended to accommodate non-static or dynamic value systems? For example, in interactive dialogue scenarios, users may introduce new values or modify existing ones during the conversation.

3.What do you perceive as the main challenges in achieving such online or incremental pluralistic value alignment? When confronted with irreconcilable deep value conflicts—for instance, those rooted in cultural or religious sensitivities—maximizing total correlation may yield compromise-like responses that lack substantive content. How do you envision PICACO, or future work building upon it, incorporating an explicit and interpretable value trade-off framework to handle such cases?

4.To what extent can the optimized meta-instruction generalize to unseen tasks or domains that were not included in the training compositions? Does the meta-instruction retain its alignment capability across out-of-distribution prompts or novel value compositions?

---

> ### Author Response · Authors · 2025-11-22
> **Thank you for your positive feedback and valuable insights. (1/3)**
>
> Thank you for your positive feedback and valuable insights, which is really important to us. We reply to each weakness (W) and question (Q) as follows. A revision of our paper has been uploaded, with the revised parts marked in blue.
>
> ---
>
> ### **W1: Lack of a theoretical framework for modeling value conflicts**
> We propose PICACO to address the *Instruction Bottleneck* challenge, in which LLMs struggle to accommodate multiple values, even when they are not in conflict, and then either overlook certain values or exhibit bias toward them.
>
> To take a step further, let us consider the case of two values, $v_1$ ad $v_2$. When the two values are fully conflicting, *e.g.*, $v_1$=*"power"* and $v_2$=*"anti-power"*, PICACO naturally struggles to reconcile them. Fortunately, such complete opposition is rare in practice.
>
> Therefore, from a mathematical perspective, we can decompose each value into a mutually compatible (shared) part and a value-specific conflicting part, analogous to a Venn diagram, *i.e.*, $v_1 = (v^s_1, v^u_1)$, $v_2 = (v^s_2, v^u_2)$.
> For example, given $v_1$=*"helpfulness"* and $v_2$=*"harmlessness"*, their shared components are $v^s_1$ = $v^s_2$ = *"helpful and harmless"*; while the conflicting components are $v^u_1$ = *"helpful but harmful"* and $v^u_2$ = *"helpless but harmless"*.
>
> Then we have:
> $$
> \text{TC}(v_1, v_2, y|x)
> = [I(v_1^s;y|x) + I(v_2^s;y|x) - I(v_1^s, v_2^s;y|x)] +
> [I(v_1^u; y|v^s_1, x) + I(v_2^u; y|v^s_2, x) - I(v_1^u,v_2^u;y|v_1^s,v_2^s,x)]
> $$
>
> $$
> = \text{TC}(v_1^s,v_2^s;y|x) + \underbrace{I(v\_1^u; y|v^s\_1, x) + I(v_2^u; y|v^s_2, x) - I(v_1^u,v_2^u;y|v_1^s,v_2^s,x)}_{\Delta\_{\text{con}}},
> $$
>
> where the first term, $\text{TC}(v_1^s,v_2^s;y|x)$, is our original optimization objective in Sec. 3.2, Eq. (1), which encourages the model to find consensus responses that capture the shared semantics of the intended values.
> The second part, $\Delta_{\text{con}}$, handles the conflicting components. Within this part, we see the first two terms help increase the correlation between $y$ and $v_1^u, v_2^u$, that is, they further absorb the semantics of each value. The last term, $I(v_1^u,v_2^u;y|v_1^s,v_2^s,x)$, captures *how* the two values, $v_1^u$ and $v_2^u$ disagree or conflict.
>
> We therefore assign a weighting hyperparameter $\lambda$ to $\Delta_{\text{con}}$, which allows us to control the trade-off of the conflicting values:
> 1. When $\lambda>0$ and increases, PICACO tends to enforce the full coverage of the semantics of $v^u_1$ and $v^u_2$. However, when $v^u_1$ and $v^u_2$ cannot be naturally reconciled, the target model may produce compromise-like responses, which are still useful in many cases (see our response to Q3). This is because $I(v_1^u,v_2^u;y|v_1^s,v_2^s,x)$ is minimized, causing the information about *how $v_1^u$ and $v_2^u$ disagree* to be lost.
> 2. When $\lambda\leq 0$ and decreases, PICACO tends to ignore or penalize any information about $v_1^u$ and $v_2^u$. In this case, our method focuses only on how to exploit the shared parts of the two values.
>
> We have included the detailed derivations and discussions in App. D.4.1.
>
> ---
>
> ### **W2: Dependence on external models for $q_\omega$**
> We have conducted an ablation study on the **external value evaluator $q_\omega$** , and tried another slightly weaker LLM, Moonshot-v1-8k [1], which underperforms on long-context [2] and user-centric [3] tasks. The results are as follows:
>
> | Variant | Confucianism | Liberalism | HH Balance | Avg.
> | -------- | -------- | -------- |-|-
> | PICACO$_{4omini}$ | *3.788* | **3.135** | *4.257*| **3.727**
> | PICACO$_{Moon}$ | **3.796** | *3.107* | 4.255 | *3.719*
> | OPRO$_{4omini}$ | 3.713 | 2.961 | **4.286** | 3.653
> | OPRO$_{Moon}$ | 3.746 | 3.027 | 4.214 | 3.662
>
> We can see **PIPACO maintains comparable performance with a weaker $q_\omega$ , and consistently surpasses the strongest baseline, OPRO, regardless of which external model is used**.
>
> Please refer to our response to Reviewer Fckv's W1.1 and App. E.2.1 of the updated pdf for more details. Thank you!
>
>
> ---
>
> ### **Q1: What are the costs of PICACO?**
> The estimated total **inference cost is $TNM\_1(2nM_2+1)+NM_2(n+1)$ generated responses**, where $T$ is the number of iterations, $N$ is the number of task prompts used for optimization, and $M_1$ and $M_2$ are the hyperparameters in Eq. (3). Here, $n$ denotes the number of responses sampled for estimating response probabilities when the target LLM is proprietary. In this paper, if the target model is GPT-3.5-Turbo, the estimated API cost should be ~$250 to align with each value composition.
>
> Please refer to our response to Reviewer 69p1's W3 and App. C.1.4 for a detailed breakdown of the cost. Thank you!

---

> ### Author Response · Authors · 2025-11-22
> **Thank you for your positive feedback and valuable insights. (2/3)**
>
> ### **Q2: How would PICACO handle dynamic values?**
> We elaborate on this question from two aspects:
>
> 1. **High data efficiency and flexibility of PICACO**. As described in our response to Reviewer 69p1's W3, PICACO requires minimal data ($N=50$ task prompts) to optimize a meta-instruction for each value composition. Moreover, benefiting from LLMs' comprehension ability, PICACO does not depend on well-established social-science value systems; it only requires that the target values can be clearly and accurately expressed in natural language (e.g., the Helpful, Honest, and Harmless principle [4]).
>
> 2. **Potential workflow for dynamic value alignment**. Based on the advantages above, it is convenient to incorporate new values into the optimization process as long as a few corresponding demonstrations are provided. Define $e^*$ as the optimized meta-instruction for the existing value composition $V$, if we want to extend it to a new value $v_{new}$:
>
>     i) *In cross-cultural, contextual, and personalized scenarios*, we ask the user to provide demonstrations, use the original $e^*$ as the the seed meta-instruction $e^{0'}$, set $V^{'}=V\cup${$v_{new}$} as the new value composition, and then conduct several additional rounds of offline instruction optimization before deploying the target LLM.
>
>     ii) *In real-time, online conversational scenarios*, there are two main challenges.
>
>     + *Users cannot pause the interaction to provide enough demonstrations*. One potential solution is to equip PICACO with RAG, retrieving demonstrations from the web or user history, or prompting a stronger external LLM to generate synthetic pseudo-demonstrations.
>     + *Instant instruction optimization introduces response latency*. Two rough ideas to mitigate this are: (a) *incremental optimization*, where optimization is performed in parallel with the conversation and the instruction is progressively updated rather than waiting for full convergence; and (b) *caching precomputed instructions*, by building a *(value, instruction)* database and retrieving the meta-instruction most relevant to $v_{new}$ as $e^0$, then performing only one fast round on top of it.
>
> We have included these discussions in App. D.4.2.
>
> ---
>
> ### **Q3: What are the main challenges in value trade-off, and how would PICACO address it?**
> 1. Our primary motivation is to tackle the issue that LLMs fail to accommodate multiple values, even when they are not conflicting, and then either miss or exhibit bias toward some values, i.e., the *Instruction Bottleneck* problem (L052).
>
> 2. We agree that, when faced with dilemmas or strongly conflicting values, PICACO may produce compromise-like responses. We have some rough ideas on how to handle this, depending on the context:
>     i) *Conflicting values carry similar weights*. Compromise-like responses are acceptable here, e,g., balancing helpfulness and harmlessness, since we care about the overall benefit of the trade-off. We plan to investigate how to further adapt PICACO to achieve Pareto-optimal alignment [5,6] across multiple values in an ICL manner.
>     ii) *Conflicting values carry different weights*. The importance of each value may depend on the context. A learned component could be used to infer a contextual weight for each intended value and apply it to the total correlation; however, this requires additional training data, introducing a new trade-off between performance and cost.
>
> We plan to explore these ideas in future work.
>
> ---
>
> ### **Q4: Can the optimized meta-instructions generalize across tasks/compositions?**
> 1. The meta-instructions of PICACO are *task/prompt-agnostic* but *tied to specific value compositions*. Once a meta-instruction $e^*$ for a value composition $V$ is optimized, it can be applied across a broad range of tasks, but embody only the values in $V$.
>
> 2. To demonstrate PICACO's generalizability on OOD tasks, we compare five meta-instructions for GPT-3.5-Turbo on the two Schwartz value compositions in this paper using an unseen benchmark, Value Portrait [7]. In addition to the tasks in the three benchmarks used by PICACO (*moral dilemmas, sentence completion*, and *viewpoint statements*), this benchmark also includes new tasks such as *creative writing* and *thread replies*. The results (overall conformity) are as follows:
>
>     | Method | Confucianism | Liberalism |
>     | -------- | -------- | -------- |
>     | Q  | 3.491 | 3.206 |
>     |Q+IF|3.522|3.184
>     |URIAL|2.648|2.726
>     |OPRO|*3.530*|*3.262*
>     |PICACO|**3.598**|**3.323**
>
>     where **PICACO consistently outperforms other baselines on unseen tasks**. This can be attribute to PICACO's *softer* optimization of meta-instructions, i.e., operating in probability space via repeated sampling, which develops more robust value conformity across diverse inputs.
>
> More details are provided in App. E.3.

---

> > ### Author Response · Authors · 2025-11-22
> > **Thank you for your positive feedback and valuable insights. (3/3)**
> >
> > We hope that our responses and additional results have addressed your concerns, and we are more than willing to answer any further questions.
> >
> > Again, we sincerely appreciate your valuable feedback and questions, which have inspired us to further explore the realm of multi-value alignment and continue improving PICACO.
> >
> > ---
> >
> > ### References
> > [1] Moonshot AI, Moonshot-v1 Documentation (https://platform.moonshot.ai/docs/introduction#generation-model-moonshot-v1), accessed 11-18-2025.
> >
> > [2] Song et al., Counting-Stars: A Multi-evidence, Position-aware, and Scalable Benchmark for Evaluating Long-Context Large Language Models, COLING 2025.
> >
> > [3] Wang et al., A User-Centric Benchmark for Evaluating Large Language Models, EMNLP 2024.
> >
> > [4] Anthropic, A General Language Assistant as a Laboratory for Alignment, arXiv:2112.00861.
> >
> > [5] Rame et al., Rewarded soups: towards Pareto-optimal alignment by interpolating weights fine-tuned on diverse rewards. NeurIPS 2023.
> >
> > [6] Yang et al., Rewards-in-Context: Multi-objective Alignment of Foundation Models with Dynamic Preference Adjustment. ICML 2024.
> >
> > [7] Han et al., Value Portrait: Assessing Language Models’ Values through Psychometrically and Ecologically Valid Items, ACL 2025.

---

> > > ### Comment · Reviewer_kUR8 · 2025-11-26
> > >
> > > The authors' response has largely addressed my main concerns regarding computational cost, method generalizability, extensibility to dynamic values, and reliance on external evaluators. The added experiments and theoretical supplements have notably strengthened the method's credibility and applicability. While the theoretical modeling of value conflicts remains somewhat preliminary and the handling of deeply irreconcilable values is still exploratory, these limitations do not diminish the method's overall contribution to real-world multi-value alignment scenarios. In summary, the authors' supplementary answers are comprehensive and rigorous, and the core issues have been substantially alleviated. I recommend that the paper be accepted by the conference.

---

> > > > ### Author Response · Authors · 2025-11-27
> > > >
> > > > We are glad that the suggested experiments and clarifications help demonstrate the effectiveness of PICACO. Yes, our supplementary theoretical framework for conflict handling and dynamic alignment is conceptually workable, and we plan to further develop both directions in future work. Thank you very much for reading our responses and for your prompt and positive feedback!

---

### Official Review · Reviewer_Fckv · 2025-10-31

**Soundness:** 3
**Presentation:** 2
**Contribution:** 2
**Rating:** 4
**Confidence:** 4

**Summary:**

The paper proposes a training-free framework (PICACO) for pluralistic alignment: instead of fine-tuning an LLM, it iteratively (i) generates and filters responses to a task using value scorers, and (ii) refines a meta-instruction so that future generations better satisfy multiple, potentially concurrent values (e.g., safety, helpfulness, style). The method is cast as maximizing a TC-like objective with two components: a value-conformity scorer $q_\omega(\cdot)$ and a redundancy-reduction scorer $q_\phi(\cdot)$, so that responses are both value-aligned and not merely surface copies. It runs as an EM-like loop over instructions and responses.

**Strengths:**

**S1: Problem framing**: the paper targets a realistic setting—multi-value / pluralistic alignment without re-training the base LLM—and gives a clear formulation of “generate → score → refine instruction” as an iterative procedure.

**S2: Simple, practical mechanism**: the two-step loop (response enhancement + instruction refinement) is lightweight, LLM-only, and can be plugged on top of existing models without extra finetuning or RL, which is attractive for third-party / resource-limited users.

**S3: Empirical coverage**: the paper evaluates on several tasks and value sets, showing consistent improvements over single-instruction baselines and demonstrating that the redundancy penalty helps avoid superficial, template-like responses.

**Weaknesses:**

**W1: Objective not fully grounded.** There is a gap between the theoretical formulation and the actual implementation of $q_\omega(\cdot)$ and $q_\phi(\cdot)$ in this paper.
 - **W1.1**. In the experiments, treating GPT-4o-mini as $q_\omega(\cdot)$ is somewhat misaligned with the theory. In the derivation, $q_\omega(\cdot)$ is presented as a variational distribution, i.e., a controllable/optimizable distribution that approximates the true posterior. Under that definition, it should be a small, explicitly parameterized model that we train ourselves, with parameters $\omega$, and that outputs a probability for each value. In practice, however, the authors use GPT-4o-mini as $q_\omega(\cdot)$, which is not “variational inference” but rather “using an off-the-shelf LLM as a value scorer.” In other words, they do not actually learn a variational distribution; they simply let GPT-4o-mini act as an oracle judge to rate how well a response satisfies the (k)-th value. This theory–implementation gap weakens the claimed probabilistic grounding: the empirical gains may come from using a strong external judge, not from the proposed variational objective.
 - **W1.2**. Redundancy detector $q_\phi(\cdot)$ is a weak spot. The whole “punish surface-level alignment” idea hinges on $q_\phi(\cdot)$ ​ being able to tell copying from genuine alignment, but the paper does not say (i) how $q_\phi(\cdot)$ ​ is trained, (ii) on what data, (iii) how robust it is across domains. If $q_\phi(\cdot)$ ​ is even slightly mis-specified, the algorithm may penalize fluent but correct responses or, conversely, fail to penalize template copying.

**W2: Ablation is incomplete.** You may need ablations such as: remove redundancy term $q_\phi(\cdot)$; fix the instruction pool (no refinement); use random instead of TC-guided instruction sampling. Without these, we can’t tell which part actually gives the lift.

**Questions:**

**Clarity of notation and definitions (Lines 194–195).**
The writing in this part is quite opaque. You state that “*s is the content where v is derived …*”, but it is unclear what **s** concretely denotes. Is **s** an in-context example or task instruction? What does it mean, operationally, that “the value (v) is derived from (s)” — does (s) *contain* value-specific textual evidence, or does it *index* the ICL example from which the value was learned? Please provide a precise definition of **s**, describe how it is obtained in practice, and, ideally, add a small running example to demonstrate how ((x, s, v)) are related.

**Contradictory objective (Lines 230–231).**
   You write that you “maximize $log q_\phi(s|x_i, y_{i,j})$ and minimize $log q_\phi(s|x_i, y_{i,j})$”. Mathematically, this is seriously self-contradictory, since it optimizes the same quantity in opposite directions. Is this a typo, or are there actually *two* different scorers involved (e.g., a positive/value-consistent scorer and a negative/redundancy scorer) that were written with the same symbol $q_\phi(\cdot)$ ? Please clarify the intended objective and explain how the same term could be both *maximized* and *minimized* in the algorithm.

---

> ### Author Response · Authors · 2025-11-21
> **Thank you for your insightful and valuable feedback. (1/3)**
>
> Thank you for your insightful and valuable feedback! We reply to each weakness (W) and question (Q) as follows. A revision of our paper has been uploaded, with the revised parts marked in blue.
>
> ---
>
> ### **W1: Gap between formulation and implementation**
> #### **W1.1: GPT-4o-mini as $q_\omega$ & unclear source of empirical gains**
> Thanks for this interesting question! We respond to it from two aspects.
>
> 1. **Is $q_\omega$ we used a valid variational distribution?**
> $q_\omega$ is incorporated in this derivation step in Eq. (14) at the beginning of App. D.3:
>
>     $$
>     \text{I}_{e}(v_k; y|x) = \mathbb{E}\_{p_e(x, y)}  \text{KL}[p_e(v_k| x, y)||q(v_k| x, y)]  + \mathbb{E}\_{p\_e(x)} \iint p\_e(v\_k, y| x) \log \frac{q(v\_k|x, y)}{p\_e(v\_k| x)} dvdy \geq \mathbb{E}\_{p_e(x)} \iint p_e(v_k, y| x) \log q(v_k|x, y) dvdy + \mathcal{H}\_{p_e}[v_k]
>     $$
>
>
>     where the inequality holds because $\text{KL}[p_e(v_k| x, y)||q(v_k| x, y)] \geq 0$. Based on this, we have:
>
>     + **As a variational distribution, $q_\omega$ is NOT required to accurately approximate the true posterior.**  The closeness of $q_\omega$ to the posterior only affects how tight the bound is, *not the validity of the inequality*, regardless of the choice of $q$.
>     + **$q_\omega$ only needs to be a valid probability distribution over $V$**, that is, $q_\omega(v|\cdot) > 0$ for any intended value and $\sum_{v \in V} q_\omega(v|\cdot)=1$; *it does not have to be a small, explicitly parameterized model that we train ourselves*. These conditions only influence the performance and efficiency of our method, but *not the validity of our mathematical derivation*.
>     + **GPT-4o-mini can be a valid probability distribution**. In this work, grounded in a well-defined value theory, e.g., Schwartz theory, $V$ is discrete, i.e., $V=$ {$v_1, v_2, ..., v_{10}$} with each $v_k$ representing one of the 10 value dimensions in Schwartz. By applying a softmax layer on top of GPT-4o-mini's scores for the 10 values, we ensure that $q_\omega(v_k|\cdot) > 0$ for any $v_k$ and $\sum_{k} q_\omega(v_k|\cdot)=1$. Note that, mathematically, **there is NO restriction on how $q_\omega$ is implemented** (e.g., whether it is a fine-tuned small model that directly outputs a valid probability distribution, or a rule-based construction corresponding to a Dirac delta distribution).
>
> 2. **The source of empirical gains**. To better understand the source of PICACO's improvement, we conducted additional ablation studies on the optimization objective and the backbone model of $q_\omega$ .
>
>     i) **Ablation on optimization objective**. With the same $q_\omega$ (GPT-4o-mini), we compared four objectives:
>
>     - PICACO: our method with the proposed **Total Correlation as the objective**.
>     - PICACO$\_{MI}$ : PICACO w/o $q_\phi$ and both redundancy reduction terms in Eq. (2), using **Mutual Information as the objective**.
>     - PICACO$_{NRP}$ : PICACO w/o both **R**esponse **P**ools described in Sec. 3.2 (L244-247) and the second redundancy reduction term in Eq. (2), directly sampling responses for each meta-instruction and computing a **TC-like, but history-independent score as the objective**.
>     - OPRO: the top-performing baseline with an **MI-like, history-independent score as the objective**.
>
>     With GPT-3.5-Turbo as the target model and GPT-4o-mini as the backbone model of $q_\omega$ , the results are as follows:
>
>     | Variant | Confucianism | Liberalism | HH Balance | Avg.
>     | -------- | -------- | -------- |-|-
>     | PICACO | **3.788** | **3.135** | *4.257*| **3.727**
>     | PICACO$_{MI}$ | 3.717 | 2.907 | 4.210 | 3.611
>     | PICACO$_{NRP}$ | *3.725* | *3.111* | 4.236 | *3.691*
>     | OPRO | 3.713 | 2.961 | **4.286** | 3.653
>
>     Clearly, with the same external $q_\omega$ , *PICACO consistently outperforms other optimization objectives*.
>
>
>     ii) **Ablation on the external value evaluator $q_\omega$** . We also tried Moonshot-v1-8k [1], a slightly weaker LLM that underperforms on spatial [2], long-context [3], and user-centric [4] tasks compared with GPT-4o-mini, for implementing $q_\omega$ . With the two different $q_\omega$ , we compared OPRO and PICACO as follows:
>
>     | Variant | Confucianism | Liberalism | HH Balance | Avg.
>     | -------- | -------- | -------- |-|-
>     | PICACO | *3.788* | **3.135** | *4.257*| **3.727**
>     | PICACO$_{Moon}$ | **3.796** | *3.107* | 4.255 | *3.719*
>     | OPRO | 3.713 | 2.961 | **4.286** | 3.653
>     | OPRO$_{Moon}$ | 3.746 | 3.027 | 4.214 | 3.662
>
>      As shown above, *PIPACO consistently surpasses the other method regardless of which external $q_\omega$  is used*.
>
>      These results together demonstrate the **improvement comes from PICACO’s novel optimization objective design, rather than from mere external signals**.
>      More experimental details are provided in App. E.1 and App. E.2.1.

---

> ### Author Response · Authors · 2025-11-21
> **Thank you for your insightful and valuable feedback. (2/3)**
>
> #### **W1.2: Unclear statement of the redundancy evaluator $q_\phi$**
> 1. **The implementation of $q_\phi$** . Due to space limitations, the details of $q_\phi$ was placed in App. C.1.2 (L1402-1460). It is based on embedding similarity computed using the jina-embeddings-v3 model [5]. Specifically, given a task prompt $x$, a response $y$, and the text observations of the intended values, $s$, we define $q_\phi(s|x,y)=1+Sim(s,y)-Sim(x,y)$.
>
> 2. **How does $q_\phi$ work?** We treat every intended value $v_k$ as a latent variable, and $s$ as the textual observations/behaviors. In practice, we set $s$ to a set of few-shot examples/demonstrations that embody each value $v_k$ (please refer to Q1 for more discussions).
>
>     The effectiveness of $q_\phi$ stems from defining surface-level alignment as *the direct copying or imitation of surface patterns from the few-shot examples (e.g., format or specific words)*, which has been observed in prior work [6][7]. Under this definition, **$q_\phi$ only needs to detect whether $y$ directly copies superficial patterns from the demonstrations in $s$**. We therefore penalize this *copy effect* as a constraint on surface-level alignment, and a simple similarity-based implementation is sufficient. In our response to W2 below, we conducted ablation studies on the redundancy terms and found that removing them led to a performance drop in PICACO$\_{NNP}$ and PICACO$\_{MI}$, indicating that $q_\phi$ indeed contributes to alignment performance.
>
>     Because we use only eight demonstrations for $s$, **severe misspecification is unlikely**. In our current implementation, the worst case is that words from these 8 demonstrations are disallowed to appear in $y$, which has only a negligible impact on fluency in practice. This is because the conformity term's weight is adjustable, preventing the redundancy terms from dominating the optimization.
>
>     More discussions have been added to App. C.1.2.
>
> ---
>
> ### **W2: Imcomplete ablation**
>
> Thanks for your valuable suggestions. Following your advice, we compared PICACO with its four variants:
> 1. PICACO$_{NNP}$ : PICACO w/o the **N**oisy **P**ool described in Sec. 3.2 (L244-247) and the corresponding second redundancy reduction term in Eq. (2). This variant is designed to assess **the effect of the noisy response pool**.
> 2. PICACO$\_{MI}$ : PICACO w/o $q_\phi$ and both redundancy reduction terms in Eq. (2), using **M**utual **I**nformation as the optimization objective. This variant is designed to evaluate **the effect of $q_\phi$** .
> 3. PICACO$_{RS}$ : PICACO w/o the second redundancy reduction term in Eq. (2), **R**andomly **S**ampling responses for all generated responses for TC calculation in each iteration. This variant is designed to assess **the effect of the *Response Enhancement Step***.
> 4. PICACO$_{Fix}$ : PICACO w/o the *Instruction Refinement Step*. Since PICACO starts with only one seed meta-instruction, we **fix**ed the instruction pool after the first iteration. This variant is designed to evaluate **the effect of the *Instruction Refinement Step***.
>
> The overall results on three value compositions, measured by GPT-4o, are as follows:
>
> | Variant | Confucianism | Liberalism | HH Balance | Avg.
> | -------- | -------- | -------- |-|-
> | PICACO | **3.788** | *3.135* | **4.257**| **3.727**
> | PICACO$_{NNP}$ | 3.691 | **3.141** | *4.217* | *3.683*
> | PICACO$_{MI}$ | 3.717 | 2.907 | 4.210 | 3.611
> | PICACO$_{RS}$ | 3.705 | 3.059 | 4.182 | 3.649
> | PICACO$_{Fix}$ | *3.719* | 2.883 | 4.158 | 3.587
>
> As demonstrated, removing the two redundancy terms, the *Response Enhancement Step*, and the *Instruction Refinement Step* results in a consistent performance drop across all three value compositions. While the absence of the second redundancy term incurred a slighter drop, it still worsened the overall alignment performance. This is probably because the responses in the noisy pool were pre-sampled and cached in limited quantities for computational efficiency.
>
> Together, these results indicate that **all the design choices you mentioned play a significant role in PICACO's superiority**.
>
> More detailed results are added to App. E.1.

---

> ### Author Response · Authors · 2025-11-21
> **Thank you for your insightful and valuable feedback. (3/3)**
>
> ### **Q1: What does $s$ in L194-195 (now L203-204) mean?**
> 1. **Theoretical definition of $s$**. We treat every intended value $v_k$ as a latent variable, and $s$ as the textual observations/behaviors. Following how traits are constructed in psychology [8], we assume that a value $v_k$ is inferred by observing, interpreting, and summarizing a large number of observed behaviors $s$. For example, after observing many people pursuing greater wealth and higher social status, we can summarize these motivational tendencies as the value of *"power"*. In practice, we define $s$ as a set of few-shot examples/demonstrations that express the value $v_k$. As stated in our response to W1.2, this helps prevent the target model from overfitting to $s$. Therefore, $s$ should consist of text that reflects the intended values but contributes minimally to responding to the task prompts.
>
> 2. **Practical implementation of $s$**. In this paper, we detail the implementation of $s$ in App. C.1.
>
>     i) For Schwartz values, we prompted the target model to elaborate on the intended values and collected its responses as $s$, since we want $s$ to be tied to the target model's inherent language distribution and the task, rather than to generic value descriptions. For example, here is an item in $s$ for the confucianism composition:
>
>         Embodying benevolence, conformity, tradition, and security means encapsulating qualities associated with kindness, compassion, and goodwill toward others (benevolence). It also involves adhering to established norms, rules, and expectations within a society or group (conformity). Additionally, it entails honoring and preserving customs, beliefs, and practices that have been passed down through generations (tradition). Lastly, embodying security means prioritizing stability, safety, and protection for oneself and others.\n\nBy embodying these values, an individual may exhibit compassion and empathy in their interactions, follow societal rules and expectations, respect and uphold cultural heritage, and prioritize safety and well-being. These values can influence how one behaves, makes decisions, and engages with others in various aspects of life, including relationships, work, and the community.\n\nOverall, embodying benevolence, conformity, tradition, and security reflects a person who values compassion, social harmony, cultural preservation, and stability in their actions and attitudes.
>
>     ii) For HH values, we set $s$ to the eight demonstration conversations, since we want the aligned model to produce responses that remain related to the specific query. Some demonstrations are shown in Fig. 6.
>
> More discussions have been added to App. C.1.1.
>
> ---
>
> ### **Q2: Is there a self-contradiction in L230-231 (now L237-238)?**
> Thank you for your attentiveness, and we apologize for any confusion caused - that was a typo. It should be *maximize* $q_\omega(v_k|x_i, y_{i,j})$ *and minimize* $q_\phi(s|x_i, y_{i,j})$, i.e., maximize the pluralistic conformity and minimize the redundancy. We have also revised the paragraph to further clarify the process.
>
> ---
>
> We hope our clarifications and additional experimental results address your concerns, and we are happy to respond to any further questions. We would sincerely appreciate it if you could read our responses and kindly reconsider the assessment of our work.
>
> ---
>
> ### References
> [1] Moonshot AI, Moonshot-v1 Documentation (https://platform.moonshot.ai/docs/introduction#generation-model-moonshot-v1), accessed 11-18-2025.
>
> [2] Xu et al., Evaluating Large Language Models on Spatial Tasks: A Multi-Task Benchmarking Study, arXiv:2408.14438.
>
> [3] Song et al., Counting-Stars: A Multi-evidence, Position-aware, and Scalable Benchmark for Evaluating Long-Context Large Language Models, COLING 2025.
>
> [4] Wang et al., A User-Centric Benchmark for Evaluating Large Language Models, EMNLP 2024.
>
> [5] Sturua et al., Jina Embeddings V3: Multilingual Text Encoder with Low-Rank Adaptations, ECIR 2025.
>
> [6] Lyu et al., Z-ICL: Zero-Shot In-Context Learning with Pseudo-Demonstrations, ACL 2023.
>
> [7] Liu et al., Making Large Language Models Better Reasoners with Orchestrated Streaming Experiences, EMNLP 2024.
>
> [8] Ye et al., Generative Psycho-Lexical Approach for Constructing Value Systems in Large Language Models, ACL 2025.

---

### Official Review · Reviewer_69p1 · 2025-11-01

**Soundness:** 3
**Presentation:** 2
**Contribution:** 2
**Rating:** 4
**Confidence:** 2

**Summary:**

This paper proposes PICACO, a black-box in-context alignment method that optimizes a meta-instruction by maximizing conditional total correlation so an LLM can express multiple, potentially conflicting human values without fine-tuning.

**Strengths:**

The method is practical. It works with black-box LLMs, requires no parameter updates.

This paper demonstrates rigorous theoretical derivation.

**Weaknesses:**

Readability: Although the paper shows the method from a theoretical view, it should also include some more intuitive, semantic explanations to improve readability.

Ablation: The authors need to test the contribution of qφ and M2 to the final results.

Cost: How much resource (GPU hours or API cost) does it take on average to align one instruction?

**Questions:**

see Weaknesses

---

> ### Author Response · Authors · 2025-11-20
> **Thank you very much for your insightful comments. (1/2)**
>
> Thank you very much for your insightful comments. Here we reply to each weakness (W) and question (Q) as follows. A revision of our paper has been uploaded, with the revised parts marked in blue.
>
> ---
>
> ### **W1: Readability**
> Thank you for the suggestion!
>
> 1. Due to space limitations, we have to place most explanations and text samples in appendix. For example, Fig. 6 in App. C.1 exemplifies $s$, i.e., the demonstration conversations, and Tables 2-4 in App. C.3 illustrate the seed meta-instructions $e^0$ used by PICACO.
>
> 2. To further improve the readability of our work, we've conducted a comprehensive review and polishing. Key changes (marked in blue) include:
>
>     i) Adding a **comprehensive notation table** of all symbols used in our mathematical expressions in App. D.1.
>
>     ii) Elaborating on the meaning of Eq. (1), our core theoretical idea, in plain language in Sec. 3.2.
>
>     iii) Incorporating concrete examples for key variables, e.g., $e$ and $x$, in Sec. 3.1, to help the understanding of our framework.
>
>     iv) Adding a running example for Algorithm 1 in App. C.1.3.
>
>     v) Providing a natural-language explanation of how the TC-based optimization works in App. D.2.
>
>     Please refer to the updated paper for more details.
>
> ---
>
> ### **W2: Missing ablations on redundancy reduction**
> We fully recognize the need to investigate the contribution of the redundancy-reduction components in PICACO.
>
> In response to the reviewers’ collective feedback, we have conducted a series of ablation studies on redundancy reduction with GPT-3.5-Turbo as the target model, including：
> 1. PICACO$_{NNP}$ : PICACO w/o the **N**oisy **P**ool described in Sec. 3.2 (L244-247) and the corresponding second redundancy reduction term in Eq. (2). This variant is designed to assess **the effect of the noisy response pool**.
> 2. PICACO$\_{MI}$ : PICACO w/o $q_\phi$ and both redundancy reduction terms in Eq. (2), using **M**utual **I**nformation as the optimization objective. This variant is designed to evaluate **the effect of $q_\phi$** .
> 3. PICACO$_{RS}$ : PICACO w/o the second redundancy reduction term in Eq. (2), **R**andomly **S**ampling responses for all generated responses for TC calculation in each iteration. This variant is designed to assess **the effect of the *Response Enhancement Step***.
> 4. PICACO$_{NRP}$ : PICACO w/o both **R**esponse **P**ools described in Sec. 3.2 (L244-247) and the second redundancy reduction term in Eq. (2), directly sampling responses for each new meta-instruction. This can be viewed as another implementation of OPRO using PICACO’s detailed settings, designed to isolate **the effect of the EM-like loop**.
>
> The overall results on three value compositions, measured by GPT-4o, are as follows:
>
> | Variant | Confucianism | Liberalism | HH Balance | Avg.
> | -------- | -------- | -------- |-|-
> | PICACO | **3.788** | *3.135* | **4.257**| **3.727**
> | PICACO$_{NNP}$ | 3.691 | **3.141** | 4.217 | 3.683
> | PICACO$_{MI}$ | 3.717 | 2.907 | 4.210 | 3.611
> | PICACO$_{RS}$ | 3.705 | 3.059 | 4.182 | 3.649
> | PICACO$_{NRP}$ | *3.725* | 3.111 | *4.236* | *3.691*
>
> As demonstrated, removing $q_\phi$ , the *Response Enhancement Step*, and the EM-like loop results in a consistent performance drop across all three value compositions. While the absence of the noisy response pool incurred a slighter drop, it still worsened the overall alignment performance. This is probably because the responses in the noisy pool were pre-sampled and cached in limited quantities for computational efficiency. Together, these results indicate that **the redundancy-reduction components play a significant role in PICACO's superiority**.
>
> The detailed results are included in App. E.1.

---

> ### Author Response · Authors · 2025-11-20
> **Thank you very much for your insightful comments. (2/2)**
>
> ### **W3: Unknown resource cost of PICACO**
> According to the implementation details described in App. C.1, PICACO's computational cost can be estimated as follows:
>
> 1. **Data**: For each value composition, we used $N=50$ task prompts as the pseudo-training set $\mathcal{X}$, and $|s|=8/16$ few-shot demonstrations for HH values and Schwartz values, respectively, to calculate redundancy.
>
> 2. **GPU**: The GPU usage only depends on $q_\omega$ or $q_\phi$ as most LLMs are queried through APIs. In this work, $q_\omega$ is implemented via GPT-4o-mini API, and $q_\phi$ through a jina-embedding-v3 model, which consumed ~3GB of GPU RAM.
>
> 3. **API usage**: Suppose the target model is proprietary and aligned via API calls.
>
>     i) For the *Response Enhancement Step*, $N \times M_1$ candidate aligned responses are sampled every iteration, and $N \times M_2$ noisy responses are sampled and cached in the first iteration.
>
>     ii) For the *Instruction Refinement Step*, in each iteration, $M_2$ new instructions are generated. Guided by each of these $M_2$ new instructions and the $M_2$ previous best instructions, another $n$ responses are sampled for each maintained aligned response to each task prompt to estimate its probability. This results in $2 \times M_2 \times N \times M_1 \times n$ generated responses per iteration. The probability of the noisy responses also requires another $N \times M_2 \times n$ responses, but only once, since they are cached and fixed.
>
>     Given the number of iterations $T=10$, if the process doesn't converge early, the maximum number of API calls needed is computed as $T \times (N \times M_1 + 2 \times M_2 \times N \times M_1 \times n) + N \times M_2 + N \times M_2 \times n=TNM_1(2nM_2+1)+NM_2(n+1)$, which amounts to \~1.5M generated responses in this paper. If the target model is GPT-3.5-Turbo, the estimated API cost should be \$\~250, depend on the detailed implementation of repeated sampling.
>
> As described above, PICACO minimizes the need for offline resources (data and GPUs) by leveraging online resources (API usage), enabling it to easily adapt to different target models (open-source or proprietary) and arbitrary value compositions. While this approach increases API usage costs, the resulting improvements in flexibility and scalability justify the investment.
>
> A more detailed account of the computational cost is provided in App. C.1.4.
>
> ---
>
> We hope our responses, additional results and the revised paper could address your concerns. We are more than willing to respond to any further questions regarding our methods and experiments.
>
> We would sincerely appreciate it if you could review our responses and kindly reconsider the assessment of our work.

---

### Official Review · Reviewer_1Ftv · 2025-11-01

**Soundness:** 3
**Presentation:** 3
**Contribution:** 2
**Rating:** 4
**Confidence:** 4

**Summary:**

The paper introduces PICACO, a training-free framework for pluralistic in-context alignment (ICA) that enables large language models to follow multiple, potentially conflicting human values simultaneously. It formulates meta-instruction optimization as conditional Total Correlation maximization between target values and responses, using an EM-like loop for response enhancement and instruction refinement. Tested across five value sets (Helpfulness, Harmlessness, HH-Balance, Confucianism, and Modern Liberalism) and multiple LLMs, PICACO consistently outperforms ICA baselines such as OPRO, URIAL, and Modular Pluralism, achieving better balance across up to eight values and improved safety robustness without finetuning.

**Strengths:**

Reframes pluralistic in-context alignment as conditional Total Correlation optimisation, offering a theoretically grounded and training-free approach that overcomes the Instruction Bottleneck in existing ICA methods.

Demonstrates rigorous empirical evaluation across five value sets and multiple LLM families (open and black-box), showing consistent improvements in alignment balance, robustness to optimisers, and enhanced safety under jailbreak tests

The paper clearly motivates the problem, explains the method with mathematical precision, and provides intuitive examples (e.g., reconciling Tradition vs. Hedonism) to illustrate nuanced value alignment

Advances the frontier of alignment research by showing that meta-instruction optimisation without finetuning can yield balanced, culturally adaptive, and safety-consistent alignment across diverse moral frameworks

**Weaknesses:**

Much of the evidence relies on LLM-as-judge (e.g., GPT-4o/DeepSeek judging outputs), which can favour certain response styles and blur whether gains reflect true value adherence versus judge preference.

The method relies on conditional Total Correlation (TC) maximisation. But there isn’t a clean ablation isolating TC from other design choices (e.g., pool construction, filtering) or comparing against alternative dependencies (e.g., MI, HSIC).

All reported comparisons are among in-context alignment methods rather than training-based ones. Even if the performance might be lower, understanding the gap is important.

**Questions:**

Can the authors add a small human evaluation?

The study only contrasts PICACO with other ICA methods. Could the authors include a lightweight SFT or DPO baseline to quantify the remaining performance gap and support the claim that PICACO offers competitive training-free alignment?

---

> ### Author Response · Authors · 2025-11-20
> **Thank you for your thoughtful reviews and suggestions. (1/2)**
>
> Thank you for your thoughtful reviews and suggestions, which have greatly inspired us in improving our work.  We reply to each weakness (W) and question (Q) as follows. A revision of our paper has been uploaded, with the revised parts marked in blue.
>
> ---
>
> ### **W1: Unknown reliability of LLM-as-a-judge**
> 1. We have employed three methods to ensure the reliability of LLM judgement:
>
>     i) **Carefully designed evaluation rubrics and templates.** We used the previously validated, widely adopted prompt templates from [1] (Figs. 9 & 10 in App. C.2). The rubrics were also sourced from well-established frameworks such as Anthropic’s RLHF principles, NVIDIA’s HelpSteer, Schwartz’s theory of basic human values, and Grice’s Maxims (App A.1).
>
>     ii) **Multi-trial sampling to minimize random errors.** We sampled nine responses for each query in the test sets and reported the average score to mitigate any bias of the judge toward certain responses, as described in App. C.2 (L1557-1559).
>
>     iii) **Multiple judge LLMs to mitigate style preference.** As mentioned in Sec. 4.2 (L321), besides GPT-4o, we also employed DeepSeek-V3.1, a high-capacity model from a different family than any other models used. As shown in Table 11, *with DeepSeek-V3.1-based $q_\omega$ , PICACO still maintains its overall superiority*.
>
> 2. To further confirm the reliability of LLM judge, following the reviewer's suggestions, **we conducted a small human study** involving the two Schwartz value compositions and five ICA methods.
> We randomly sampled 10% of GPT-3.5-Turbo's responses, and each response was independently scored twice by three human annotators. With both the inter-rater agreement and the human-LLM agreement measured using Krippendorff’s alpha, the results are as follows:
>
>     | Value Composition | Method | Human Score |
>     | -------- | -------- | -------- |
>     ||Q+IF+CoT|2.928|
>     |**Confucianism**|URIAL+Sum|3.517|
>     |(Inter-rater: $\alpha$=0.94)|Modular|3.458|
>     |(Human-LLM: $\alpha$=0.93)|OPRO|*3.585*|
>     ||PICACO|**3.674**|
>     ||||
>     ||Q+IF+CoT|2.794|
>     |**Liberalism**|URIAL+Sum|2.987|
>     |(Inter-rater: $\alpha$=0.79)|Modular|*3.145*|
>     |(Human-LLM: $\alpha$=0.88)|OPRO|3.029|
>     ||PICACO|**3.186**|
>
>     where PICACO still outperforms most baselines, and there is strong alignment between the LLM judge (GPT-4o-2024-08-06) and human annotators. More details of the human study have been added to App. E.2.2.
>
> **All these practices indicate the results and conclusions are less likely to be affected by the LLM judges' stylistic preferences.**
>
> ---
>
> ### **W2: Missing ablations on TC**
> Thank you for the valuable suggestion! We have conducted a series of ablations on PICACO's design factors, including a variant, PICACO$_{MI}$, that uses Mutual Information instead of Total Correlation as the objective.
>
> Here is the comparison between PICACO and PICACO$_{MI}$ on three of the value compositions in our paper:
>
> | Variant | Confucianism | Liberalism | HH Balance
> | -------- | -------- | -------- |-
> | PICACO | **3.788** | **3.135** | **4.257**
> | PICACO$_{MI}$ | 3.717 | 2.907 | 4.210
>
> As shown in the results, replacing TC with MI results in a consistent performance drop across all three value compositions, which suggests **the effectiveness or TC as PICACO's optimization objective**. We speculate that this is because TC takes the redundancy in responses into considerations via additional $q_\phi$, thus implicitly increasing the density of value-relevant content.
>
> We have also conducted **five additional ablation studies on other design choices** of PICACO. Please refer to our responses to Reviewer 69p1’s W2 (the noisy pool, the *Response Enhancement Step*, and the EM-like loop) and Reviewer Fckv’s W1-1 ($q_\omega$) and W2 (the *Instruction Refinement Step*) for more results and analyses.
>
> The experimental details of the ablation studies are provided in App. E.1 and App. E.2.1.

---

> ### Author Response · Authors · 2025-11-20
> **Thank you for your thoughtful reviews and suggestions. (2/2)**
>
> ### **W3: Lack of a training-based baseline**
> Thank you for your insightful suggestion and we also agree that training-based baselines can help better illustrate the advantages of our method.
>
> 1. **Following the reviewer's advice, we fine-tuned GPT-3.5-Turbo** using OpenAI's supervised fine-tuning (SFT) API. Here we compared four methods:
>
>     i) Q+IF: GPT-3.5-Turbo w/ Q+IF (in this paper, ICL),
>
>     ii) Q+IF$_{SFT}$ : Fine-tuned GPT-3.5-Turbo w/ Q+IF (new, SFT),
>
>     iii) PICACO: GPT-3.5-Turbo w/ PICACO (in this paper, ICL), and
>
>     iv) Q+IF$_{4o}$ : GPT-4o-mini w/ Q+IF (new, pseudo-tuning-based).
>
>     For Q+IF$\_{SFT}$, we used samples from the *FULCRA* [2] and *HH-RLHF* [3] datasets as training data for Schwartz and HH value compositions, respectively. Moreover, since GPT-4o-mini is officially claimed to be a stronger substitute for GPT-3.5-Turbo [4], we included Q+IF$_{4o}$ for HH values only.
>
>     | Method | Confucianism | Liberalism | HH Balance | Helpfulness | Harmlessness
>     | -------- | -------- | -------- | - | - | -
>     | Q+IF |*3.306*|*2.728*|4.082|*4.247*|4.032
>     | Q+IF$_{SFT}$ |3.258|-|4.121|4.210|4.081|
>     | Q+IF$_{4o}$ |-|-|*4.193*|4.241|*4.083*|
>     | PICACO |**3.788**|**3.135**|**4.257**|**4.287**|**4.173**
>
>     As shown above, **PICACO consistently outperforms the training-based baseline across all value compositions**, suggesting that our ICA method is superior to naive SFT methods in multi-value settings.
>
> 2. **The reasons we didn't include a training-based baseline before**.
>
>     i) The primary reason is the significant **challenge of obtaining sufficient training data for arbitrary value compositions**, due to pluralistic human values and heterogeneous real-world needs, as noted in Sec. 1 (L38 & L45-46).
>
>     ii) Moreover, **insufficient fine-tuning data leads to degraded performance**. We tried to collect *all* data from *FULCRA* that embody at least three values in the *Confucianism* composition (*Benevolence, Conformity, Tradition, Security*), but were able to obtain only 1.5k samples; there are very few qualified training samples for the *Liberalism* composition in *FULCRA* (<50). The sample size for *HH-RLHF* was set to 2k for consistency. As shown above, Q+IF$_{SFT}$ even underperforms the untuned Q+IF, possibly because the large model tends to overfit to limited samples.
>
> Therefore, a major advantage of PICACO is its **exceptional data efficiency** (only eight curated demonstrations are required to align LLMs with each value composition).
>
> More details of these SFT baselines have been added to App. E.4 of the updated paper.
>
> ---
>
> For Q1 and Q2, please refer to our responses to W1 and W3, respectively.
>
> We hope our clarifications and additional experimental results address your concerns, and we are happy to respond to any further questions. We would sincerely appreciate it if you could read our responses and kindly reconsider the assessment of our work.
>
> ---
>
> ### References
> [1] Lin, et al., The Unlocking Spell on Base LLMs: Rethinking Alignment via In-Context Learning, ICLR 2024.
>
> [2] Yao, et al., Value FULCRA: Mapping Large Language Models to the Multidimensional Spectrum of Basic Human Value, NAACL 2024.
>
> [3] Anthropic, Training a Helpful and Harmless Assistant with Reinforcement Learning from Human Feedback, arXiv:2204.05862.
>
> [4] OpenAI, GPT-3.5 Turbo Documentation (https://platform.openai.com/docs/models/gpt-3.5-turbo), accessed 11-16-2025.

---

### Author Response · Authors · 2025-11-27
**A revised version of our submission including all additions has been uploaded.**

We thank all reviewers for their thoughtful and constructive feedback. We fully understand that reviewers and ACs are quite busy, and we sincerely appreciate any further feedback they may have during the discussion period.

Following the reviewers' advice, we have uploaded a revised PDF that now includes **all supplementary discussions, derivations, implementation details, and additional experimental results** referenced in our responses. All section and line references in our responses have been adjusted to match the revised version.

We hope these additions and clarifications facilitate a fair reassessment of our work, and we are also more than happy to address any further concerns or questions. Again, we are grateful for the time and effort the reviewers have devoted to our submission.

---

### Author Response · Authors · 2025-12-01
**Rebuttal Summary for AC (1/2)**

Dear AC,

We would like to express our sincere thanks to all reviewers, and especially to the ACs, for their time and efforts. In hopes of reducing your additional workload caused by the unexpected situation, we have provided a summary of **reviewers' feedback** along with **our solutions and responses**.

---

### **Reviewers' feedback**
1. Reviewers **consistently emphasized several core strengths** of PICACO, including:
    - *A well-motivated formulation of a practical problem* (1Ftv, 69p1, Fckv, kUR8)
    - *Intuitive plausibility demonstrated through examples and formulations* (1Ftv, kUR8)
    - *Strong theoretical contribution through rigorous derivations* (1Ftv, 69p1, kUR8)
    - *Methodological effectiveness across diverse settings* (1Ftv, Fckv, kUR8)

    In addition, reviewer 1Ftv acknowledged that our *evaluation is comprehensive and rigorous*, while reviewer Fckv noted that our *simple and practical mechanism is well-suited for resource-limited scenarios*.

2. The **concerns** mainly center on the experiments, more specifically, on **a more fine-grained evaluation of PICACO's effectiveness and efficiency** rather than on flaws in its design:
    - *Incomplete ablation study leaving the contribution of each component unclear* (1Ftv-W2, 69p1-W2, Fckv-W1/W2)
    - *Unknown resource cost* (69p1-W3, kUR8-Q1)
    - *Dependence on an external value evaluator* $q_\omega$ (Fckv-W1.1, kUR8-W2)
    - *Uncertain reliability of LLM-as-a-judge for evaluation* (1Ftv-W1/Q1)
    - *Absence of a training-based baseline* (1Ftv-W3/Q3)

    In addition, reviewer Fckv raised questions regarding the *design rationale and detailed implementation* of two key components of PICACO, i.e., the value evaluator $q_\omega$ and the redundancy evaluator $q_\phi$ . Reviewer kUR8 posed a series of questions concerning PICACO's *generalizability and extensibility*.

---

### **Our solutions & additions**
In our responses, we comprehensively addressed each reviewer's concerns by:

- conducting **additional experiments for an in-depth, fine-grained evaluation** of PICACO, yielding **48 additional overall results** across diverse ICA methods, experimental settings, and value compositions (38 automatic + 10 human);

- providing **detailed examples, explanations, derivations and discussions** for specific questions; and

- uploading **a revised paper with comprehensive additions**, with the revised parts marked in blue.

With GPT-3.5-Turbo as the target model and GPT-4o as the judge model, our **main efforts in additional experiments** include:

1. **A series of ablation studies** *(1Ftv-W2, 69p1-W2, Fckv-W1/W2; detailed in Appendix E.1)*

    We evaluate PICACO with its five variants, each created by removing one component, and analyze the resulting change in improvement (Improv.) over the raw target LLM. This allows us to assess the effect of:

    i) the noisy response pool for the second redundancy reduction term (PICACO$_{NNP}$, Improv. $\downarrow$ 10.8%),

    ii) the redundancy evaluator $q_\phi$ (PICACO$_{MI}$, Improv. $\downarrow$ 28.6%),

    iii) the Response Enhancement Step (PICACO$_{RS}$, Improv. $\downarrow$ 19.2%),

    iv) the Instruction Refinement Step (PICACO$_{Fix}$, Improv. $\downarrow$ 34.5%), and

    v) the EM-like loop (PICACO$_{NRP}$, Improv. $\downarrow$ 8.9%).

    We demonstrate that **each design choice** questioned by the reviewers indeed **significantly contributes to PICACO's superior performance**, and that **the proposed Total Correlation optimization objective plays an important role in PICACO's empirical gains**.

2. **An analysis of the implementation of** $q_\omega$ *(Fckv-W1.1, kUR8-W2; details are added to App. E.2.1)*

    We use a slightly weaker LLM, Moonshot-v1-8k, as a substitute for GPT-4o-mini when implementing the value evaluator $q\_\omega$ , resulting in PICACO$\_{Moon}$. With the two different $q_\omega$, we compared the strongest baseline, OPRO, and PICACO. The average scores across three value compositions, *i.e., Confucianism, Liberalism, and HH Balance*, are: PICACO$\_{4omini}$ (**3.73**) vs. OPRO$\_{4omini}$ (3.65), and PICACO$\_{Moon}$ (**3.72**) vs. OPRO$\_{Moon}$ (3.66).

    These results demonstrate that **PIPACO consistently surpasses the baselines regardless of which $q_\omega$ is used**, showing robustness to the external $q_\omega$ .

---

> ### Author Response · Authors · 2025-12-01
> **Rebuttal Summary for AC (2/2)**
>
> 3. **An SFT baseline** *(1Ftv-W3/Q3; full experimental settings and analysis are in App. E.4)*
>
>    We fine-tune GPT-3.5-Turbo using OpenAI's supervised fine-tuning (SFT) API. Here we compare four methods:
>
>     i) Q+IF: GPT-3.5-Turbo w/ Q+IF (in the original submission, ICL),
>
>     ii) Q+IF$_{SFT}$ : Fine-tuned GPT-3.5-Turbo w/ Q+IF (new, SFT),
>
>     iii) PICACO: GPT-3.5-Turbo w/ PICACO (in the original submission, ICL), and
>
>     iv) Q+IF$_{4omini}$ : GPT-4o-mini w/ Q+IF (new, pseudo-tuning-based).
>
>     The results on all five value compositions are as follows:
>
>     | Method | Confucianism | Liberalism | HH Balance | Helpfulness | Harmlessness
>     | -------- | -------- | -------- | - | - | -
>     | Q+IF |*3.31*|*2.73*|4.08|*4.25*|4.03
>     | Q+IF$_{SFT}$ |3.26|-|4.12|4.21|4.08|
>     | Q+IF$_{4omini}$ |-|-|*4.19*|4.24|*4.08*|
>     | PICACO |**3.79**|**3.14**|**4.26**|**4.29**|**4.17**
>
>     where **PICACO consistently outperforms the tuning-based baseline across all value compositions**, suggesting that our ICA method is superior to naive SFT methods in multi-value settings.
>
> 4. **A human study comparing GPT-4o with human judges** *(1Ftv-W1/Q1; more details are added to App. E.2.2)*
>
>     We conduct a small human study involving the two Schwartz value compositions and five ICA methods. We randomly sampled 10% of GPT-3.5-Turbo's responses, and each response was independently scored twice by three human annotators. With both the inter-rater agreement and the human-LLM agreement measured using Krippendorff's alpha, the results are as follows:
>
>     | Value Composition | Method | Human Score |
>     | -------- | -------- | -------- |
>     ||Q+IF+CoT|2.93|
>     |**Confucianism**|URIAL+Sum|3.52|
>     |(Inter-rater: $\alpha$=0.94)|Modular|3.46|
>     |(Human-LLM: $\alpha$=0.93)|OPRO|*3.59*|
>     ||PICACO|**3.67**|
>     ||||
>     ||Q+IF+CoT|2.79|
>     |**Liberalism**|URIAL+Sum|2.99|
>     |(Inter-rater: $\alpha$=0.79)|Modular|*3.15*|
>     |(Human-LLM: $\alpha$=0.88)|OPRO|3.03|
>     ||PICACO|**3.19**|
>
>     where **PICACO still outperforms most baselines**, and there is **strong alignment between the LLM judge (GPT-4o-2024-08-06) and the human annotators** (human-LLM Krippendorff's alpha $\geq$ 0.88).
>
> 5. **OOD evaluation of PICACO and OPRO** *(kUR8-Q4; more details are added to App. E.3)*
>
>     We compare meta-instructions from five ICA methods on the two Schwartz value compositions in this paper using an external benchmark, *Value Portrait*. In addition to the tasks in the three benchmarks used by PICACO, this benchmark also includes OOD tasks such as *creative writing* and *thread replies*. The results are as follows:
>
>     | Method | Confucianism | Liberalism |
>     | -------- | -------- | -------- |
>     | Q  | 3.49 | 3.21 |
>     |Q+IF|3.52 |3.18
>     |URIAL|2.65 |2.73
>     |OPRO|*3.53*|*3.26*
>     |PICACO|**3.60**|**3.32**
>
>     where **PICACO generalizes better than other baselines on OOD tasks**, exhibiting more robust value conformity across diverse inputs.
>
> In addition to these experiments, we provide various clarifications, including a **detailed breakdown of PICACO's resource cost covering data, GPU, and API usage** in App. C.1.4.
>
> ---
>
> As all reviewers noted, our research *excels in problem formulation, theoretical foundation, and methodological effectiveness*; **most concerns centered on the unclear role of each component and the method’s dependence on the external value evaluator, rather than on any fundamental flaw in our approach**. We believe the **additional explanations and results above have adequately addressed these doubts and further reinforced our design and claims**.
>
> Please note that reviewer kUR8, who initially raised the most questions about PICACO, has acknowledged that we have *"largely addressed their concerns and substantially alleviated the core issues"* and recommended that our paper be accepted (on Nov 25, before the large-scale leakage).
>
> We sincerely hope that you could review this summary and assist the other reviewers in making **a fair reassessment of our revised work**. Again, thank you for your important efforts in supporting the stability of the ICLR community!
>
>
> Best regards,
>
> The authors

---

### Meta-Review · Area_Chair_Qevg · 2025-12-27

**Summary:**

Reviewers were generally negative (scores of 4, 4, 4, 8), with concerns largely around (1) the motivation behind the proposed complex method (e.g., why the use of total correlation for this particular task? what's special about value alignment beyond generic instruction optimization?) and questions about related ablations, and (2) the dependence on an external model (GPT-4o-mini) that's generally stronger than the LMs used for evaluation, which is not a realistic scenario because one could generally just use GPT-4o-mini directly. In response, the authors ran an extensive set of ablation experiments and also tried a slightly weaker external model (notably still reasonably strong) that still worked well.

The rebuttal was quite extensive. I suspect that the reviewers would have raised their scores slightly post-rebuttal, though the overall scores are still borderline. As the reviewers noted, the direction of this paepr is interesting and it has the potential to be impactful. I recommend that the authors revise their work to incorporate the feedback and all of their rebuttal experiments, and then resubmit.

**Reviewer Concerns:**

The ablations were helpful, though they were fairly localized and the overall motivation is still unclear, as is the use of external models.

**Reviewer Scores:**

The 4s might have changed to 5s; not sure there would be large changes.

---

### Decision · Program_Chairs · 2026-01-26

Reject